# MGL S3 Chimeric Enzyme Drives Apoptotic Death of EGFR-Dependent Cancer Cells through ERK Downregulation

**DOI:** 10.3390/ijms232112807

**Published:** 2022-10-24

**Authors:** Nikolay Bondarev, Karina Ivanenko, Elmira Khabusheva, Timofey Lebedev, Ilya Manukhov, Vladimir Prassolov

**Affiliations:** 1Research Center for Molecular Mechanisms of Aging and Age-Related Diseases, Moscow Institute of Physics and Technology, Institutsky Lane 9, 141700 Dolgoprudny, Russia; 2Engelhardt Institute of Molecular Biology, Russian Academy of Sciences, Vavilov Street 32, 119991 Moscow, Russia; 3Center for Precision Genome Editing and Genetic Technologies for Biomedicine, Engelhardt Institute of Molecular Biology, Russian Academy of Sciences, Vavilov Street 32, 119991 Moscow, Russia; 4Federal State Budgetary Educational Institution of Higher Education, Moscow State University of Food Production, Volokolamskoye Highway 11, 125080 Moscow, Russia

**Keywords:** ERK signaling, methionine, gefitinib

## Abstract

Methionine dependence of malignant cells is one of the cancer hallmarks. It is well described that methionine deprivation drives cancer cells death, both in vitro and in vivo. Methionine gamma-lyase (MGL) isolated from different species or obtained by genetic engineering can be used for effective methionine depletion. In this work, we show that MGL S3, a genetically engineered protein comprised of MGL from *Clostridium sporogenesis* fused to epidermal growth factor (EGF)-like peptide, reduces, in vitro, the number of cancer cells of four different origins—neuroblastoma, lung, breast, and colon cancer. We reveal that MGL S3 is more toxic for neuroblastoma SH-SY5Y and lung cancer H1299 cells compared to MGL tetani, and causes cell death by the induction of apoptosis. In addition, the observed death of cells treated with MGL S3 is accompanied by the prominent downregulation of ERK activity. By the analysis of transcriptomic data of more than 1500 cancer cell lines and patient samples, we show that the high expression of four genes from the methionine metabolism pathway (*AHCY*, *CBS*, *DNMT3A*, and *MTAP*) is associated with poor prognosis for breast cancer and neuroblastoma patients. Additionally, cells of these origins are characterized by a high correlation between EGFR dependency and *DNMT3A*/*CBS* expression. Finally, we demonstrate the ability of MGL S3 to enhance the sensitivity of H1299 cells to EGFR inhibition with gefitinib.

## 1. Introduction

Altered metabolism is one of the cancer hallmarks [1]. Cancer cells depend on the availability of external L-methionine (methionine). Methionine is an essential sulfur-containing amino acid (SAA) that is catabolized and recycled in a series of metabolic reactions named the methionine metabolism cycle. Different organisms, ranging from archaea to plants, but not human, possess a unique enzyme, methionine gamma-lyase (MGL), which directly degrades SAA to alpha-keto acids, ammonia, and volatile thiols. The methionine metabolism cycle of human cells includes methionine adenosyl transferases (MATs) MAT1A (liver-specific) and MAT2A/MAT2B (extrahepatic) [2], methionine synthase (MTR), S-adenosyl-L-homocysteine hydrolase (SAHH), nicotinamide N-methyltransferase (NMNT), and several other enzymes [3]. In human cells, methionine is required for the production of L-cysteine, which is critical for protein synthesis, glutathione, polyamine synthesis, and S-adenosylmethionine (SAM) formation. Methionine conversion into SAM results in the release of the methyl group, which is utilized further by methyltransferases. Thus, there is an interconnection between methionine and another crucial hallmark of tumors—epigenetic deregulation [4].

The majority of cancer cells of different origin, including breast, colorectal cancer, and leukemic cells, are methionine dependent compared to non-malignant cells that require only homocysteine in the growth media to proliferate normally [5,6,7]. Dietary methionine restriction, which reduces but does not completely eliminate methionine, results in significant tumor regression in cell line- and patient-derived xenograft mouse models of colorectal cancer [5]. Elimination of methionine induces cytotoxicity on a vast panel of human cancer cell lines, including colon cancer cells [8]. It has been shown that tumors, including cancer of the neuroendocrine system such as glioblastomas, medulloblastoma, and neuroblastomas, are much more sensitive to methionine starvation than normal corresponding tissues [9]. In several pre-clinical trials, the safety and tolerability of a methionine restriction diet was demonstrated [5,10,11].

There are several described mechanisms of methionine dependency of cancer cells, including mutations in 5-methyltetrahydrofolate-homocysteine methyltransferasegene *MTR* (present in approximately 8% of cancer samples), deletions in methyl-thioadenosine phosphorylase gene *MTAP* (found in approximately 15% of cancer samples including melanoma and pancreatic cancer [12,13]), and low levels or complete absence of methionine synthase [3,14,15]. Interestingly, cancer stem cells exhibit the elevated activity of enzymes within the methionine metabolism, most notably an upregulation of *MAT2A* expression and elevated activity of the corresponding enzyme [16,17].

In addition to diet, there are several approaches for targeting the methionine metabolism and dependency of cancer cells, including enzymes and small molecules. Enzymatic depletion of methionine can be achieved by MGL or pyridoxal 5′-phosphate. Various wild-type MGLs with different activity, off-target effects, and side toxicity have been obtained by direct purification from bacteria, protozoa, and plants. For anticancer application the recombinant protein, MGL can be obtained by the delivery of the corresponding gene into *Escherichia coli* [18,19,20]. Neuroblastoma cell lines are highly sensitive to recombinant MGL [9].

In addition, recombinant fusion proteins consisting of MGL linked to the different peptides are promising for the selective and effective elimination of cancer cells in vitro and in vivo [21,22]. Although methionine deprivation alone possesses significant antitumor activity, it also exerts significant synergistic effects on the in vivo reduction of tumor growth when combined with widely used cancer therapies, including the chemotherapeutic drug 5-fluorouracil for colorectal cancer [23] and doxorubicin and cisplatin for lung cancer [24].

In this study, we have aimed to describe the sensitivity of cancer cells to a genetically engineered MGL fusion protein selective for EGFR-positive cells (Application for a patent of the Russian Federation No. 2022113614 dated 20 May 2022 “MGL-S3 chimeric enzyme—methionine gamma lyase fused with the S3 domain of the VGF protein from Vaccinia virus, a method for producing MGL-S3 and an antitumor drug based on this enzyme”. Authors: Bondarev N. A., Bazhenov S. V., Pokrovsky V. S., Okhrimenko I. S., Baranova A. V., Manukhov I. V.) by obtaining ERK-KTR reporter cancer cell lines with lentiviral vector delivery. S3 peptide [25] was added to the C-terminus of the recombinant MGL protein from *Clostridium sporogenesis*, defining the selection of a genetically-engineered enzyme for EGFR-positive cells. We used four cell lines—breast, lung, colon cancer, and neuroblastoma—as a model for the characterization of cell properties after the methionine reduction, and tested novel drug combinations with prominent anticancer activity.

## 2. Results

### 2.1. MGL S3 Treatment Reduces Cancer Cell Number and Induces Apoptosis

According to the different studies, cancer cell methionine addiction varies dramatically (reviewed in [2]). For example, breast cancer SK-BR-3 and neuroblastoma SK-N-SH cell lines are methionine dependent, whilst breast HCC70 and pancreas PANC1 cancer cells are not.

Given that the response of cancer cells to bacterial-derived and genetically engineered MGL is not defined by their origin [2,8,24], we have assessed the sensitivity of four human cancer cell lines (H1299, SH-SY5Y, SK-BR-3, and HCT116) to MGL tetani and novel recombinant protein MGL S3, obtained by genetic engineering, consisting of MGL and EGF-like peptide.

First, cells were exposed to various concentrations (1–5 U/mL) of MGL tetani and MGL S3 and incubated for 72 h, and their viability was measured by XTT assay (Figure 1a). The most prominent reduction in cell viability was observed for SH-SY5Y and H1299—2.5 U/mL of MGL tetani and MGL S3 reduced their viability more than twice. On the contrary, 2.5 U/mL and a twice higher concentration (5 U/mL) of either MGL tetani or MGL S3 caused not more than a 10% reduction of HCT116 and SK-BR-3 cells viability.

Based on the cell viability determined by the XTT assay, the IC50 value of MGL was calculated for each cell line. Among these cell lines, the lowest IC50 of MGL tetani was determined for SH-SY5Y—0.8 U/mL, and the lowest IC50 of MGL S3 for H1299—1.1 U/mL (Table 1).

As the metabolic activity measured by Cell Proliferation tests do not always correspond to the number of viable cells, we have additionally calculated the total cell number of cells by processing photos obtained by fluorescence microscopy in Cell Profiler. Briefly, we have counted cell nuclei marked with DAPI (nuclei stain). We have observed at least a double reduction in cell number in all four studied cell lines in response to 2.5 U/mL of MGL tetani and MGL S3 (Figure 1b).

Next, we have calculated IC50 values for MGL S3 and MGL tetani, showing that non-small lung cancer H1299 cells are the most sensitive to MGL S3 (IC50 = 0.6 U/mL), while the breast cancer SK-BR-3 cell line is the most resistant (IC50 = 3.1 U/mL) (Table 2).

Non-small lung cancer H1299 and colon cancer HCT116 are at least two-times more sensitive to MGL S3 compared to MGL tetani, highlighting the increased dependency of these cells on EGFR. The sensitivity of SH-SY5Y neuroblastoma cells to MGL S3 and MGL tetani is likely to be EGFR independent as we observed equal toxicity of these two agents. In contrast, MGL tetani is more cytotoxic to SK-BR-3 than MGL S3. Notably, we have observed massive nuclei damaging in SK-BR-3 cells treated with MGL S3 (Appendix A). On the other hand, MGL tetani reduced the cell number without causing any significant alteration of nuclei and cell shape.

Interestingly, according to previously published data [8], SK-BR-3 cells had one of the highest MGL-BL929 IC50 value (1.87 U/mL) and HCT116 the lowest (0.18 U/mL) among more than 30 tested cancer cell lines. Unfortunately, data for SH-SY5Y and H1299 were not presented in that study. Notably, the cytotoxicity of methionine gamma lyase against normal skin fibroblasts was twice as low compared to the most MGL-unsensitive H1993 lung adenocarcinoma cell line [8]. In addition, in our previous work we showed the good tolerability of MGL from Clostridium sporogenes in a mouse model [24].

Collectively, these findings indicate that SH-SY5Y, SK-BR-3, H1299, and HCT116 cancer cells are highly susceptible to recombinant MGL, which is in good agreement with previously published studies.

### 2.2. Effect of MGL Treatment on Apoptosis and BCL2 Expression

Next, we assessed the ability of MGL tetani and MGL S3 to induce apoptosis of cancer cells. We selected two cell lines for further studies: H1299—the most sensitive to MGL S3 and SK-BR-3—the most resistant, according to the cell number calculation assays. To determine whether methionine-depleted cell culture media can lead to the induction of apoptosis in cancer cells, we incubated H1299 and SK-BR-3 cells with 2.5 U/mL of MGL tetani and MGL S3 for 48 h. We found 35–40% of apoptotic SK-BR-3 cells treated with MGL tetani and S3 (Figure 2a). MGL tetani induced only minor apoptosis in H1299 cells and MGL S3 induced the death of 20% of cells (Figure 2a and Appendix A). Interestingly, on microphotographs we observed significant nuclei damage and shrinkage in SK-BR-3 cells, whilst the nuclei of H1299 cells remained round and non-injured (Figure 1b).

Next, we measured the mRNA level of anti-apoptotic protein BCL2 in the MGL-treated cell. BCL2 is upregulated in the majority of cancer cells of different origins and can impact the survival of cells treated with various anticancer drugs [26]. We have found upregulation of *BCL2* mRNA in SK-BR-3 cells exposed to MGL S3, thus driving our attention to the induced ability of these cells to survive for a relatively long time under methionine deprivation and, possibly, adapt to MGL S3, as we also have detected a three-fold higher IC50 value for MGL S3 compared to MGL tetani in the cell counting assay. Indeed, in multiple studies, an elevated level of BCL2 was linked to cancer cells acquiring resistance to various chemotherapeutic molecules [27,28]. Comparatively, the *BCL2* mRNA level dramatically dropped in H1299 cells in response to MGL tetani. MGL S3 caused even more prominent downregulation of *BCL2* in these cells. The observed decrease of *BCL2* mRNA in response to MGL tetani and MGL S3 is consistent with the results of the apoptosis assay obtained for H1299 cells, as the MGL S3 induces more prominent cell death compared to MGL tetani, along with the stronger induction of *BCL2*. On the other hand, both MGL proteins caused a similar apoptosis rate (approximately 40%) in SK-BR-3 cells, while *BCL2* expression was affected differently. As about, 50% of SK-BR-3 cells survive upon treatment with MGL S3 (Figure 1a), but have altered shape and damaged nuclei and high Annexin V level (Appendix A and Figure 2a); we suppose that these cells could survive long term due to the compensatory activation of BCL2. Thus, combinational therapeutic approaches are needed for the effective simultaneous targeting of several pro-survival mechanisms of cancer cells. Overall, these findings demonstrate that various cancer cells respond with cell number reduction and apoptosis induction to methionine depletion caused by exposure to MGL tetani and MGL S3.

### 2.3. ERK Activity Is Downregulated in MGL S3 Treated Cells

To further investigate the consequences of cancer cell exposure to MGL, we performed ERK activity analysis. Having demonstrated the implication of ERK in the growth factor-related resistance of neuroblastoma cells to RTK-inhibitors [29], we then investigated ERK activity in SH-SY5Y, SK-BR-3, HCT116, and H1299 to understand the implication of this kinase in the sensitivity and adaptation to MGL. The establishment of ERK-KTR reporter cell lines, as described in [29,30] allowed us to quantify ERK activity in individual live cells. The validation of the ERK activity reporter was recently described by us [31].

By quantification of the ERK activity by CellProfiler, we demonstrated the reduction in kinase activity in all tested cell lines incubated with MGL S3 for 48 h (Figure 3a,b). The most prominent effect was observed in HCT116 cells, with approximately half reduction in ERK activity by 1–5 U/mL MGL S3.

Notably, all tested concentrations of MGL S3 (from 1 to 5 U/mL) caused almost the same effect on the downregulation of ERK activity in H1299, HCT116, SH-SY5Y, and SK-BR-3 cells, showing a sufficiency of 1 U/mL to inactivate ERK, and also indicating that methionine deprivation alone cannot completely block cancer cell survival molecular machinery.

The level of ERK downregulation by MGL tetani in HCT116 and H1299 cells was comparable to that driven by MGL S3. In contrast, ERK activity was affected only mildly (less than 20%) or remained completely unaffected in SK-BR-3 and SH-SY5Y cells, respectively, upon MGL tetani treatment. All in all, MGL S3 effectively and significantly reduced ERK activity in all studied cell lines.

### 2.4. Sensitivity to MGL Does Not Depend on the Expression Level of Methionine Metabolism Genes

As a next step in the elucidation of the molecular mechanism defining MGL sensitivity and further proceeding to the establishment of promising therapeutic options based on methionine depletion, we have compared the expression of KEGG methionine metabolism genes in two groups of cells—highly and poorly sensitive to MGL-BL929. IC50 values for MGL-BL929 were obtained from [8] and the transcriptomic data of 26 cell lines were obtained from the CCLE dataset, deposited at hgserver1.amc.nl.

The sensitive group (IC50 > 1.26 U/mL) contained 10 cell lines, including HCT116, and the resistant group (IC50 < 0.64 U/mL) contained 10 cell lines, including SK-BR-3. Surprisingly, the comparison of the expression of 16 methionine metabolism genes in two groups revealed no significant difference (Figure 4a) underlaying the existence of additional cellular mechanisms of MGL sensitivity.

Additionally, we have studied four datasets containing information of the overall survival of patients diagnosed with breast (Clynes, *N* = 104), neuroblastoma (Versteeg, *N* = 88), non-small cell lung (Bild, *N* = 106), and colon (Smith, *N* = 232) cancer, and their mRNA sequencing data. We revealed that none of the 17 methionine metabolism genes had prognostic relevance for patients with lung and colon cancer. On the contrary, high expression of each of nine genes (poor prognosis: *MARS2*, *AHCY*, *SRM*, *DNMT1*, *MTR*, *MTAP*, *DNMT3A*, *CBS*, *AMD1*; good prognosis: *MAT2B*) was significantly associated with the prognosis of neuroblastoma patients and five (poor prognosis: *AHCY*, *CBS*, *DNMT3A*, *DNMT3B*, *MTAP*; good prognosis: *MAT2B*) for breast cancer patients (Figure 4b). Whereas more than 80% of the listed genes were markers of poor prognosis. Among these prognosis-associated genes, four genes, *AHCY*, *CBS*, *DNMT3A*, and *MTAP* (*p*-value < 0.05), were common markers of poor prognosis for neuroblastoma and breast cancer.

As MGL S3 was designed for the precise targeting of EGFR-positive cancer cell lines, we decided to further study the interconnection between EGFR and methionine metabolism. The existence of altered metabolism in cancer cells resistant to EGFR inhibition was previously described as one of the death escape mechanisms [32,33,34]. We hypothesized that EGFR-dependent cells can also have specific methionine metabolism signatures. By performing a combined analysis of cancer cell line dependency data from DepMap and the CCLE transcriptomic data of 1000 cell lines of 22 origins, we have clustered cell types according to their dependency on EGFR knockout and the expression of 16 genes listed in methionine metabolism KEGG. The clustering of cell lines based on the Pearson correlation coefficient between EGFR dependency score and methionine metabolism gene expression highlighted the existence of three major clusters (Figure 5).

We define rhabdoid (kidney and other soft tissues malignant tumor), gastric, neuroblastoma, breast, ovarian, and thyroid cancer cells (first cluster) as the most co-dependent on EGFR and methionine cycle genes. The second cluster contained lung cancer and colon/colorectal cancer cells with variable correlation coefficients for the studied parameters. Finally, the third cluster was formed by bone cancer cells that are characterized with the inversed correlation between EGFR dependency and the expression of metabolism genes. 

Interestingly, earlier described genes associated with poor prognosis for neuroblastoma and breast cancer patients (Figure 4b)—*CBS* and *DNMT3A*—were placed in one cluster, whilst the other two genes—*MTAP* and *AHCY*—were placed in the second cluster. In addition, EGFR dependency correlated with the majority of genes from the first cluster in rhabdoid, gastric, neuroblastoma, breast, ovarian, and thyroid cancer cells. Low, or the absence, of correlation between EGFR dependency and expression is shown for *CBS* and *SRM* genes in rhabdoid cancer, for the *MTR* gene in gastric cancer and for the *SRM* gene in rhabdoid and neuroblastoma cancer cells. Genes placed in the second cluster have variable correlations with EGFR dependency across the cancer cell lines from both clusters, and thus they should be studied separately for each cancer type. Notably, MGL S3, but not MGL tetani, caused ERK downregulation in neuroblastoma (SH-SY5Y) and breast cancer (SK-BR-3) cells, that represent cancer types that have a strong correlation between EGFR dependency and methionine metabolism genes expression.

Thus, an elevated level of methionine metabolism gene expression is present in EGFR-dependent cells, and these cells can possess a higher sensitivity to methionine depletion and EGFR-inhibition or downregulation.

### 2.5. MGL S3 Synergizes with Gefitinib in Reducing the Viability of H1299 Cells

Several EGFR-targeting therapeutic agents are currently used for patients with EGFR-positive lung cancer, including receptor tyrosine kinase inhibitors such as afatinib, dacomitinib, erlotinib, gefitinib, and osimertinib. To define the alteration of sensitivity of cancer cells to one of the EGFR inhibitors, gefitinib, in response to the methionine depletion, we started with the assessment of EGFR mRNA level in H1299 and SK-BR-3 cells treated with MGL S3 and MGL tetani for 48 h. We observed dramatic downregulation of *EGFR* (more than five times) in response to MGL S3 in both cell lines (Figure 6a). MGL tetani also slightly reduced *EGFR* mRNA levels in H1299, but not in SK-BR-3 cells (Figure 6a). Simultaneously, *EGF* mRNA was upregulated in MGL tetani-treated H1299 cells, pointing to the possible existence of a compensatory mechanism affecting the EGFR/EGF signaling pathway.

Finally, we incubated H1299 cells with EGFR inhibitor gefitinib in the presence of either MGL tetani or MGL S3 and assessed their viability on day 3. An amount of 20 µM of gefitinib was used in this analysis, and this concentration of the drug reduced the viability of cells by approximately 20% (Figure 6b). MGL S3, but not MGL tetani, enhanced the cytotoxicity of the EGFR inhibitor gefitinib, thus leading to a 4× reduction of H1299 cell viability (Figure 6b). This effect can partially depend on the ability of MGL tetani to upregulate *EGF* expression, thus leading to autocrine EGFR-signaling stimulation and the survival of cells. Here we demonstrate that methionine deprivation in combination with EGFR inhibition can effectively inhibit the growth of H1299 cancer cells.

## 3. Discussion

Methionine dependency is likely to be one of the universal vulnerabilities of tumors of different origins, thus targeting the methionine metabolism pathway either with the elimination of methionine or by affecting the activity of methionine metabolism enzymes, such as MAT2A, is a promising anticancer therapeutic approach. Fusion proteins with MGL activity and selectivity to cancer cells can be obtained by genetic engineering for the selective targeting of cancer cells harboring specific surface markers. Within the last 20 years, the differential impact of MGL on cancer, but not normal cell death, has been demonstrated. However, the mechanism of the high sensitivity of cancer cells to this enzyme remains poorly studied. In addition, it is unclear how to distinguish patients that can benefit from this therapeutic approach. Moreover, targeting the metabolic function of the cell is often not enough for the complete elimination of the tumor in vivo, as some clones can survive long-term, driving relapses.

Within the current study, we used cell models to demonstrate the efficacy of MGL S3 for the inhibition of cancer cell growth and showed that it is accompanied by ERK downregulation. By the analysis of several publicly available datasets, we have shown that the sensitivity of cells to MGL does not depend on the levels of expression of 16 methionine metabolism genes. Aimed mainly at the examination of MGL S3 activity in vitro, we also revealed the novel crosstalk between EGFR signaling and methionine metabolism existent in several types of malignant cells. We have shown that, based on the correlation between EGFR dependency (measured by EGFR knockout, DepMap [35]) and the expression of 16 methionine metabolism genes, the cancer cell lines of 22 different origins can be divided into two major clusters. Among the most dependent cell lines, we have found breast cancer and neuroblastoma, but not lung and colon. Notably, the high expression level of the majority of methionine metabolism genes, as we found, are markers of poor prognosis for patients with breast cancer and neuroblastoma. However, we found no prognostic relevance of these genes for lung and colon cancer patients. Among KEGG methionine metabolism genes, we have revealed five poor prognostic markers, including *AHCY*, *CBS*, *DNMT3A*, and *MTAP*, for neuroblastoma and breast cancer patients. The implication of adenosylhomocysteinases, which mediates the conversion of S-adenosylhomocysteine (methylation inhibitor), encoded by corresponding gene *AHCY*, in the progression of cancer is still not obvious. High expression of *AHCY* is associated with MYCN-amplified neuroblastoma and contributes to the enhanced proliferation of cancer cells [36,37]. AHCY is also implicated in the regulation of cell cycle and DNA damage through the MEK/ERK signaling pathway and p53 in breast cancer cells [38]. Along with the up-regulated *AHCY*, MYCN-amplified neuroblastoma cells have elevated levels of the *CBS* gene, encoding cystathionine beta synthase. In addition, CBS contribution into the development of tumors and proliferation of malignant cells has been described for chronic myeloid leukemia [39] and breast [40], ovarian [41], and colon [42] cancer. Overexpression of DNA-methyltransferases, including *DNMT3B*, at both the mRNA and protein level has been described for several malignancies, including prostate, lung, breast, and colorectal cancer; glioblastoma; and leukemia [43,44,45,46,47]. The revealed interconnection between EGFR signaling and methionine metabolism in cancer cells was further utilized in the current study. As approximately 10% of studied cancer cells survive even under high concentrations (up to 5 U/mL) of MGL S3, we decided to test its ability to work with the known EGFR inhibitor gefitinib. Receptor tyrosine kinases, including EGFR, are the most attractive anticancer targets. However, EGFR inhibition may result in an activation of the compensatory pro-survival signaling in tumors, including antiapoptotic protein BCL2 upregulation in lung cancer cells [27]. We have shown that MGL S3, but not MGL tetani, enhances the anti-proliferative activity of gefitinib, causing the death of H1299 cells. Thus, the revealed downregulation of BCL2 in MGL S3-treated H1299 cells could be involved in the combined action of these two compounds.

The establishment of stable reporter cell lines by the lentiviral delivery of the reporter system allowed us to determine cell number and measure ERK activity in the same cells. Previously, we have shown the implication of ERK activity in the response of cancer cells to targeted therapy [29,30,48], and within this study, we have demonstrated the ability of MGL S3 to inhibit ERK. In addition, a reporter cell line based on SH-SY5Y cells was previously applied by us to study the implication of ERK activity in the differentiation of neural cells [49]. Thus, reporter cell lines obtained by lentiviral delivery are widely applicable in cancer research and can serve as a useful approach for simultaneous registration of several parameters of the cells in response to drugs, such as cell number, ERK activity, size, and morphology.

Moreover, we have described a new unexpected link between EGFR dependency and methionine metabolism gene expression in cancer cell lines. A cluster of cancer cell lines, consisting rhabdoid (kidney and other soft tissues malignant tumor), gastric, neuroblastoma, breast, ovarian, and thyroid cancer cells, that are co-dependent on EGFR and methionine, was revealed. Moreover, four genes from the methionine metabolism gene set, such, *AHCY*, *CBS*, *DNMT3A*, and *MTAP*, are markers of poor prognosis for neuroblastoma and breast, but not for colon and lung, cancer patients. Finally, we have shown that EGFR inhibition with gefitinib synthesize with MGL S3 in H1299 cells. Thus, the simultaneous targeting of EGFR and methionine metabolism could be a promising anticancer strategy.

## 4. Materials and Methods

### 4.1. Cell Lines and Cellular Assays

Neuroblastoma SH-SY5Y (SK-N-SH clone, without MYCN amplification), breast cancer SK-BR-3 (adenocarcinoma with HER2 overexpression), non-small cell lung carcinoma H1299 (with homozygous partial deletion TP53), colon cancer HCT116 (KRAS mutant), and CTH, human embryonic HEK293T (modified with SV40 large T antigen) cells were maintained at 37 °C and 5% CO_2_ in cell culture medium supplemented with 10% fetal bovine serum (Gibco, ThermoFisher Scientific, Waltham, MA, USA), 2 mM L-glutamine (Gibco, ThermoFisher Scientific, USA), 100 units/mL penicillin, 100 μg/mL streptomycin (Gibco, ThermoFisher Scientific, USA), and 1 mM sodium pyruvate (Sigma-Aldrich, St. Louis, MO, USA). RPMI-1640 (Gibco, ThermoFisher Scientific, USA) cell culture medium was used for SH-SY5Y and SK-BR-3 cells and DMEM (Gibco, ThermoFisher Scientific, USA) for H1299, HCT116, and HEK293T cells. Cell lines were provided by the Heinrich–Pette Institute at the Leibniz Institute for Experimental Virology as a gift and were routinely checked for mycoplasma contamination.

For cell viability assessment with Cell Proliferation assay XTT (Invitrogen, Waltham, MA, USA) 5000 cells were plated in TC-treated flat-bottom clear 96-well plates (TPP) 24 h before the MGL addition. The final volume of cell culture media containing cells and drugs was 100 µL. After 3 days of incubation, XTT compounds were added to the wells for 4 h and the absorption signal was measured with Multiscan FC (ThermoFisher Scientific, USA).

For apoptosis/necrosis measurement with Annexin V and Propidium Iodide, 20,000 cells were plated in TC-treated flat-bottom clear 24-well plates (TPP) 24 h before the MGL addition. After 3 days of incubation, adherent cells were dissociated from the plates with trypsin, washed with phosphate buffer saline, and stained with Annexin V-FITC (Invitrogen, ThermoFisher Scientific, USA) in 100 µL of Annexin staining buffer (0.02 mM HEPES, 0.14 M NaCl, 2.5 mM CaCl2) for 10 min. Prior to measurement on a BD LSRFortessa Flow Cytometer (BD, USA), 300 µL of ice-cold buffer containing propidium iodide was added (Sigma-Aldrich, USA).

### 4.2. Lentiviral Particles Production and Titration

An amount of 10^6^ HEK-293T cells were plated on 100 cm^2^ Petri dishes 24 h before the transfection. Calcium transfection was used to deliver plasmids into the HEK-293T cells. Cell culture media containing lentiviral particles was collected from HEK-293T cells 24 and 48 h after the transfection with plasmids, coding structural elements of the lentivirus (as described at http://www.lentigo-vectors.de, accessed on 16 October 2022), plasmid coding G protein (envelope) of the vesical stomatitis virus VSV, and a vector plasmid containing a gene of interest. To obtain ERK-KTR cell lines for ERK quantification, pLentiCMV Puro DEST ERKKTRClover plasmid (Addgene, #59150) was used, which encoded the ERK docking domain ELK, nuclei localization site, and nuclei extraction site (with phosphorylation sites) fused to fluorescent protein mClover.

### 4.3. Plasmid Construction, MGL Expression and Purification

The *Escherichia coli* strain BL21(DE3): F-ompT hsdSB gal dcm (DE3) (Novagen) was used for *Clostridium tetani* MGL and MGL-S3 gene expression. Bacteria were grown in standard conditions and selected with kanamycin (30 μg/mL) after transformation. *E. coli* BL21(DE3) transformed with pET-mgl-tet and pET-mgl-S3 was used for MGL biosynthesis. Plasmids pET-mgl-S3 were received from original pET-mgl-Sporog, using Gibson Assembly. Primers for cloning were chemically synthesized according to the corresponding amino acid residues VGF52-67 (RCSHGYTGIRCQAVVL) [25] and fused to the C-terminal end of megL in the previously constructed pET-mgl-Sporog plasmid. As a control, a MGL from *Clostridium tetani* we used, which showed the same effects as MGL from *Clostridium sporogenesis* to cancer cells and had a close kinetic property [50].

Enzyme synthesis was performed as described in [24]. To purify the MGL S3 enzyme, 200 g of cells (stored at −70 °C) were thawed and resuspended in 10 mM potassium phosphate buffer with 1 mM EDTA, 0.1% DTT, and 0.01 mM; pyridoxalphosphate (pH 7.2; buffer A) at a ratio of 1:5. The cell biomass was destructed using an M-110P microfluidizer with the addition of 0.2 mM of protease inhibitor PMSF (Thermo Scientific, USA). Cell lysate was centrifuged, and supernatant was heat-treated and centrifuged with 20% ethanol. The obtained supernatant was loaded onto a Q-Sepharose column pre-equilibrated with buffer. For endotoxin removal, the column was washed with buffer A, Triton X-114, and 5% isopropanol, according to the manufacture protocol. With anion-exchange chromatography with KCl elution, the main peak was eluted from the column by 0.3 M KCl. The collected fractions were precipitated by sulfate ammonia in two steps: 35% of sulfate ammonia waste proteins were precipitated and enzyme was collected with the supernatant; it was besieged again in 50% ammonia sulfate and precipitated. The MGL-S3 solution was desalinated and transferred to PBS buffer containing 0.01 mM pyridoxalphosphate, pH 7.2, by gel filtration using a HiPrep 26/10 desalting column. The final product was subjected to sterilizing filtration. The sterile solution was bottled in dark-glass vials, freeze dried (ALPHA 1–4 LD), and stored at 2–8 °C. For MGL, the tetani purification protocol was slightly changed in the sulfate ammonia step. The percentage of sulfate ammonia was increased from 35% and 50% to 55% and 80%. The activity of the purified enzymes was measured as described in [24]. PBS (pH 7.2) was used as a working buffer instead of 100 mM potassium phosphate buffer (pH 8.0). The determined activity of purified enzymes were 10–12 U/mg for MGL-tetani and 7–9 U/mg for MGL-S3.

### 4.4. Quantification of ERK Activity with ERK-KTR Reporter

We used two ERK-KTR cell lines (H1299 and SH-SY5Y) that were previously established and described in [29,30] and similarly obtained two additional ERK-KTR cell lines (SK-BR-3 and HCT116) to measure ERK activity in live individual cells. ERK-KTR translocation reporter was introduced into the target cells by lentiviral transduction and selected with FACS (BD Aria, USA). Cells were imaged with a Leica DMI8 automated microscope at 10× magnification. The nuclei of the cells were stained with 500 ng/mL Hoechst-33342 for 30 min before imaging. For each drug/combination of drugs, we imaged three wells and two fields in each well. Imaging of the cells was performed in two channels—461 nm (for Hoechst) and 520 nm (for mClover). Illumination correction, segmentation of nuclei and cells, and calculation of cytoplasm to nucleus ratios of individual cells (C/N ratio) corresponding to ERK activity were performed in CellProfiler.

### 4.5. RNA Isolation and Real-Time qRT-PCR

Total RNA was isolated with Trizol reagent (Applichem) and concentration and purity were assessed with NanoDrop ND-1000 (ThermoFisher Scientific, USA). An amount of 5000 ng of RNA was used for the synthesis of complementary DNA with a RevertAid First Strand cDNA Synthesis Kit (ThermoFisher Scientific, USA). cDNA was desolved in 500 µL of nuclease-free water (ThermoFisher Scientific, USA) and stored at −20°C.

Real time quantitative polymerase chain reaction (qPCR) was performed in three replicates using white hard-shell 96-well PCR plates and CFX96 Touch Real-Time PCR Detection System (Bio-Rad) according to the following protocol: 1 cycle—95 °C, 10 min; 40 cycles—95 °C, 10 s; 57 °C, 15 s; 72 °C, 15 s, followed by melt-curve analysis. For qPCR reaction, 5 µL of cDNA and 200 nM of specific primers were desolved in nuclease-free water and mixed with 3 µL of 5X qPCRmix-HS SYBR to a final volume of 15 µL per well. Specific primers used in this study: BCL2_pr1 TGAACTGGGGGAGGATTGTG; BCL2_pr2 CGTACAGTTCCACAAAGGCA; EGFR_pr1 AGGAGAGGAGAACTGCCAGAA; EGFR_pr2 TCTCGGAATTTGCGGCAGAC; GAPDH_pr1 GAGCCCGCAGCCTCCCGCT; GAPDH_pr2 GCGCCCAATACGACCAAATC. Cycle threshold values for target genes were normalized to endogenous glyceraldehyde-3-phosphate dehydrogenase gene (*GAPDH*).

### 4.6. RNA-Sequencing Data Analysis

We used four datasets, containing RNA sequencing data of samples obtained from patients diagnosed with neuroblastoma (GSE16476, Versteeg, *N* = 88) [51], non-small cell lung (GSE3141, Bild, *N* = 106) [52], breast (GSE42568, Clynes, *N* = 104) [53], and colon (GSE17538, Smith, *N* = 232) [54] cancer, and one dataset with cell lines RNA sequencing data (GSE36133, CCLE, *N* = 917) [55] downloaded from “R2: Genomics Analysis and Visualization Platform (http://r2.amc.nl, accessed on 10 October 2022)”. Dependency data (for EGFR) were obtained from CRISPR screen (DepMap, depmap.org) [35].

### 4.7. Statistical Analysis

Statistical analysis was performed using GraphPad Prism 9 software. The types of tests are defined in the description of the figures along with the number of replicates.

## 5. Conclusions

MGL S3, a novel methionine gamma lyase, obtained by genetic engineering protein comprised of MGL from Clostridium sporogenesis fused to epidermal growth factor (EGF)-like peptide, reduces cell number, induces apoptosis, and downregulates ERK activity in neuroblastoma, lung, breast, and colon cancer cells in vitro. An elevated level of the methionine metabolism gene expression signature is present in EGFR-dependent cells, such as rhabdoid (kidney and other soft tissues malignant tumor), gastric, neuroblastoma, breast, ovarian, and thyroid cancer cells. Four genes (*AHCY*, *CBS*, *DNMT3A*, and *MTAP*) from the methionine metabolism gene set are markers of poor prognosis for neuroblastoma and breast cancer patients. MGL S3 downregulates EGFR mRNA level in breast and lung cancer cell lines, and the EGFR inhibitor gefitinib enhances the sensitivity of H1299 cells to MGL S3. Thus, a novel genetically-engineered protein, MGL S3, could be used for the precise targeting of EGFR-dependent cancer cells, and combined with EGFR inhibitors.

## Figures and Tables

**Figure 1 ijms-23-12807-f001:**
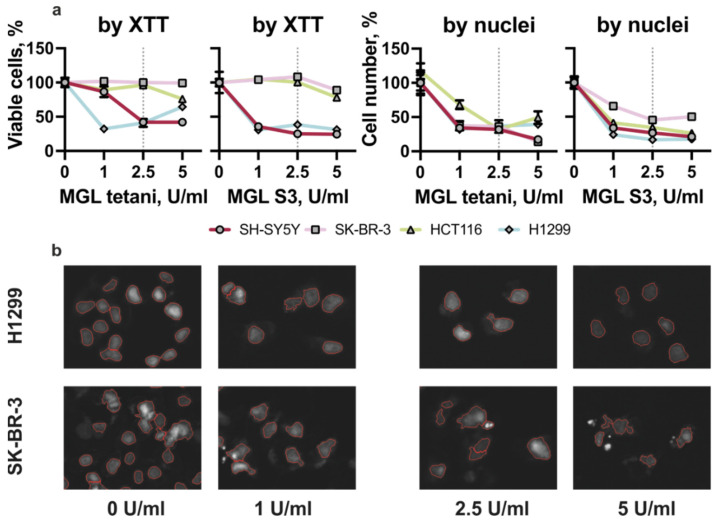
MGL S3-induced apoptosis in SK-BR-3 and H1299 cancer cells. (**a**) Viability of SH-SY5Y (circle), SK-BR-3 (square), HCT116 (triangle), and H1299 (rhombus) cancer cells incubated with 0 to 5 U/mL MGL tetani (left) or MGL S3 (right) for 72 h. The percentage of viable cells was determined with the Cell Proliferation Kit II (by XTT) and by nuclei calculation in CellProfiler (by nuclei). Data are presented as a mean, with SEM calculated for three independent replicates. (**b**) Photographs of H1299 and SK-BR-3 cells incubated with 0–5 U/mL MGL S3, processed with CellProfiler (red line shows the results of nuclei segmentation analysis).

**Figure 2 ijms-23-12807-f002:**
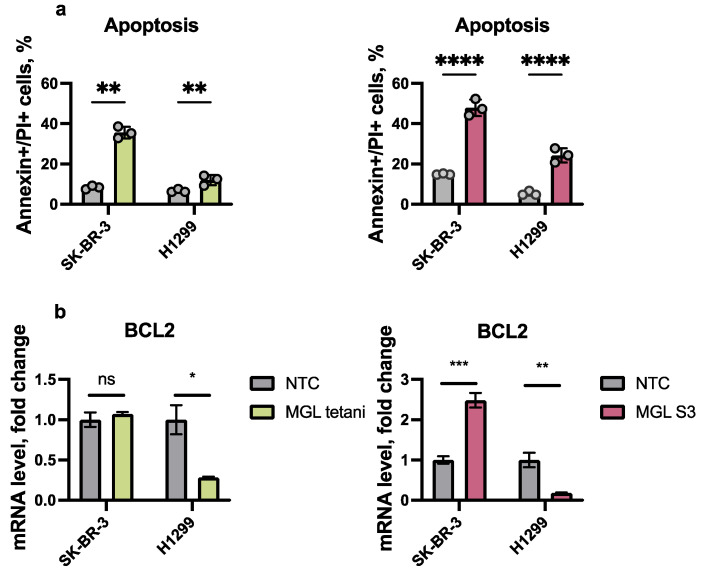
Apoptosis and BCL2 mRNA level. (**a**)The percentage of apoptotic cells in the culture of SK-BR-3 and H1299 cancer cells incubated with 2.5 U/mL MGL tetani (left panel) or MGL S3 (right panel) was determined with Annexin V-FITC staining and propidium iodide (PI) staining. (**b**) BCL2 mRNA level in SK-BR-3 and H1299 cells treated with MGL S3 and MGL tetani. Data are presented as means with the SEM of three biological replicates. Results of the two-way ANOVA (with no correction) test are represented as stars. * *p* < 0.05, ** *p* < 0.01, *** *p* < 0.001, **** *p* < 0.0001, ns—non significant. NTC—none-treated control.

**Figure 3 ijms-23-12807-f003:**
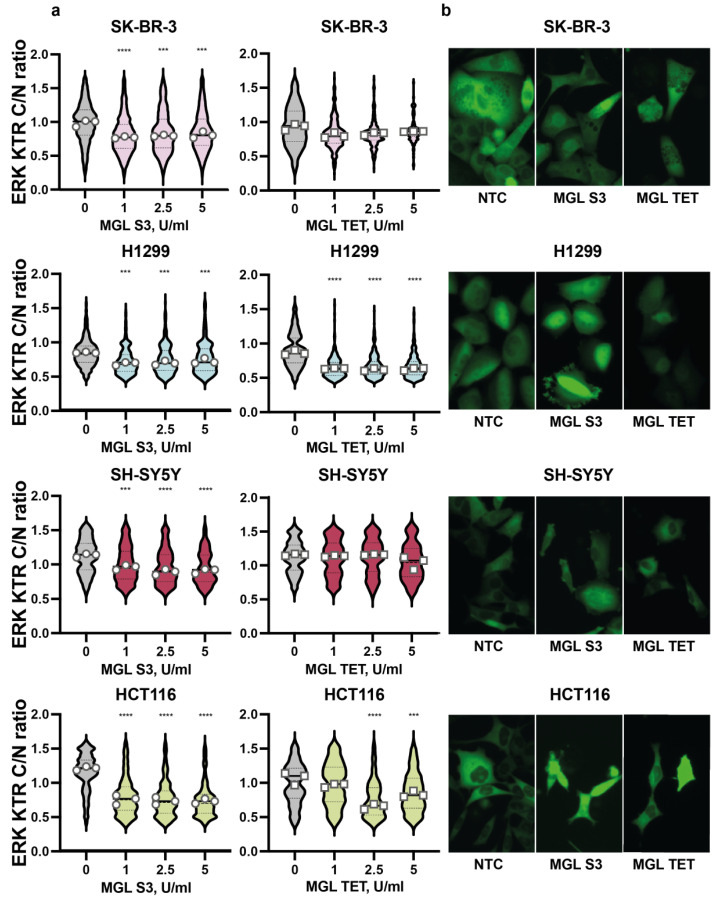
ERK activity in cancer cells treated with 1–5 U/mL MGL tetani and MGL S3. (**a**) Violin plots showing ERK activity in SK-BR-3 and H1299 cells treated with 0–5 U/mL MGL S3 and MGL tetani for 48 h were obtained by the analysis of ERK-KTR-mClover localization (cytoplasm/nucleus) in individual cells. Data are presented as a distribution and mean of three replicates (one field per well). Result of the Mann–Whitney test is indicated: *** *p* < 0.001, **** *p* < 0.0001. Variation of ERK activity exceeding 20% compared to control (0) was taken as meaningful. Values of individual replicates are indicated as circles (MGL S3) or squares (MGL tetani). (**b**) Microphotographs of SK-BR-3, H1299, SH-SY5Y, and HCT116 cells treated with 0 (left) and 2.5 U/mL MGL S3 (middle) or MGL tetani (right).

**Figure 4 ijms-23-12807-f004:**
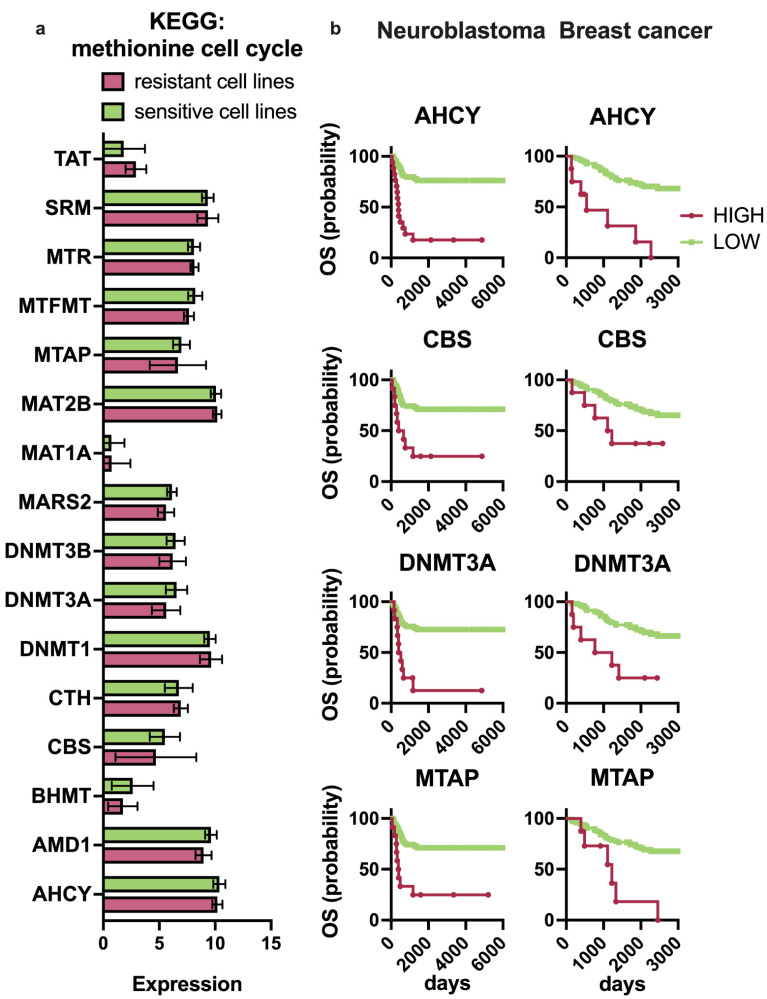
Methionine metabolism cycle genes in human cancers. (**a**) Bars representing the expression of methionine metabolism genes (KEGG) resistant to MGL-BL929 (red) and sensitive (green) cell lines. Cells were divided into two groups according to their sensitivity to MGL-BL929. (**b**) Kaplan–Meier curves for neuroblastoma (*N* = 88) and breast cancer (*N*=104). Patients’ overall survival probability (OS) based on the expression level of AHCY, CBS, DNMT3A, and MTAP. Green curve refers to the survival of patients with low expression of an individual gene, red—to high.

**Figure 5 ijms-23-12807-f005:**
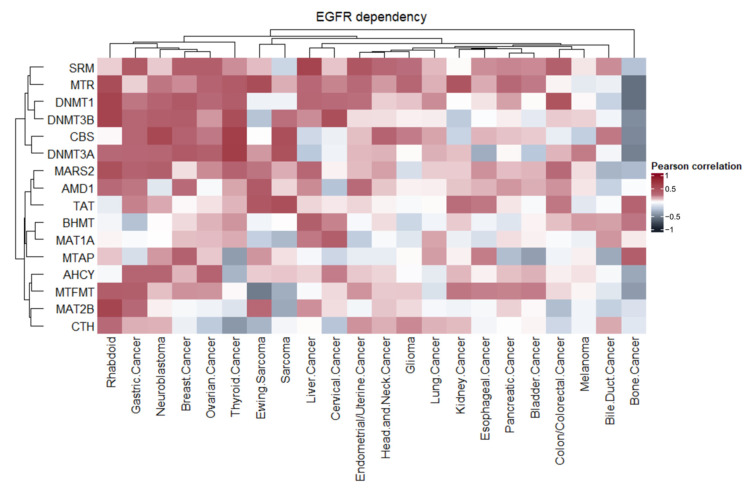
Clustering of human cancer cell lines based on the correlation coefficient of EGFR dependency score and methionine cycle gene expression. Color represents Pearson correlation coefficient, ranged from −1 (blue) to +1 (red). Pearson correlation was calculated for the gene expression level and EGFR dependency (from DepMap) of cancer cells of different origin. Clusters were obtained from the hdbscan library.

**Figure 6 ijms-23-12807-f006:**
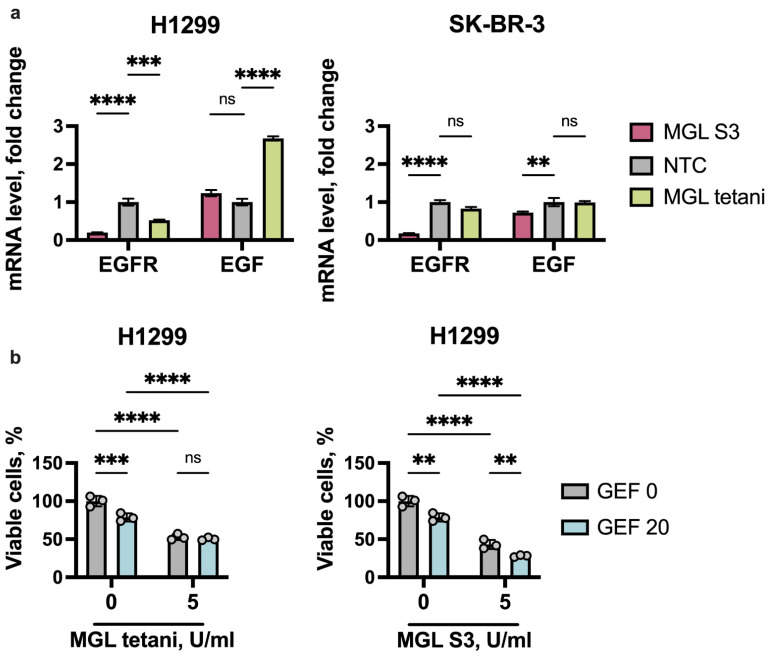
MGL S3 enhances the sensitivity of H1299 cells to the EGFR inhibitor gefitinib. (**a**) Relative mRNA level of *EGFR* and *EGF* genes in H1299 and SK-BR-3 cells incubated with MGL tetani and MGL S3 for 24 h. (**b**) The viability of H1299 cells incubated with 5 U/mL of MGL tetani (left panel) or MGL S3 (right panel) in combination with 20 µM gefitinib (GEF) was determined by XTT assay. Data are presented as a mean with the SEM of three replicates. Results of ANOVA test are represented as stars. ** *p* < 0.01, *** *p* < 0.001, **** *p* < 0.0001, ns—non significant.

**Table 1 ijms-23-12807-t001:** IC50 values (U/mL) of MGL tetani and MGL S3 by XTT assay.

Cell Line	IC50 (U/mL), MGL Tetani	IC50 (U/mL), MGL S3
SH-SY5Y	0.8	2.9
SK-BR-3	>5	>5
H1299	0.98	1.1
HCT116	>5	>5

**Table 2 ijms-23-12807-t002:** IC50 values (U/mL) of MGL tetani and MGL S3 by cell nuclei count.

Cell Line	IC50 (U/mL), MGL Tetani	IC50 (U/mL), MGL S3
SH-SY5Y	0.8	0.8
SK-BR-3	0.9	3.1
H1299	1.8	0.6
HCT116	2.2	1.1

## Data Availability

Not applicable.

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
