# Peer review of "MGL S3 Chimeric Enzyme Drives Apoptotic Death of EGFR-Dependent Cancer Cells through ERK Downregulation"

_ijms, 2022, doi:10.3390/ijms232112807_

Round 1
Reviewer 1 Report
In their manuscript Khabusheva et al. showed that the genetically engineered protein L-methionine gamma-lyase (MGL) S3 reduces in vitro the number of neuroblastoma, lung, breast and colon cancer cells. The authors showed that tumor cell death takes place through the induction of apoptosis. Moreover, the research group investigated the mechanisms underlying the ability of MGL S3 to inhibit ERK activity and to enhance the cytotoxicity of gefitinib, an EGFR inhibitor, in H1299 cells.
However, a few observations have to be taken into account:
1) Even if the question is original and well-defined, and to some extent it is relevant for the field, the manuscript is not very clear, and it contains several mistakes that have to be corrected:
a) Results section (page 2, line 95): the authors stated that experiments were done on four human cancer cell lines – HT29, SH-SY5Y, SK-BR-3, and HCT116, but further data are shown for H1299 (neuroblastoma cell line), instead of HT29 (colon cancer cell line);
b) Results section, Figure 2 legend (page 5, line 158): the staining fluorescent reagent used is PI, and not PE;
c) Materials and Methods section (page 14, line 425): the unit measure for wave length ”nm”, and not ”nM”.
2) The study design has several methodological inaccuracies. The methods are not clearly presented and materials are not sufficiently detailed that the research might be reproduced. For example, no information is given for the manuscript reader regarding:
a) The human cancer cell lines used for the in vitro studies, like histological type, where from they were obtained (which cell culture collection, or if they were a gift, and from whom);
b) No details are given in the “Materials and Methods” section for cell cultures, cell media used, cell cytotoxicity assays (except that a XTT kit was used), such as type of plates used (96-wells, flat?), volumes, number of cells/well;
c) No details are given in the “Materials and Methods” section for the flow-cytometry assay regarding the Annexin-FITC/ PI double staining for assessing the apoptotic events, like volumes, number of cells/tube.
d) The Real-time qRT-PCR assay misses also the technical details, excepting the specific primers sequences, and no reference is included.
Therefore, the manuscript’s results are not reproducible based on the details given in the methods section, and the analyses are not performed with the highest technical standards.
3) The cytotoxicity technique used was the XTT colorimetric assay. The authors should perform an interference test of the compounds used for cell treatments with the viability dye, in the absence of cells, to see if a potential interference of the tested compounds with the dye might occur for the highest concentrations.
4) The results provide a certain advancement of the current knowledge and presented in a structured manner, but the data are not interpreted appropriately and consistently throughout the manuscript.
Author Response
Dear Reviewer,
We appreciate your thoughtful comments. We believe that with your help we have improved the quality of our manuscript, described our findings and ideas more clear and precise, as well as made manuscript’s results reproducible by expanding the "Material and Methods" section.
1)
a) We apologize for this error. In this manuscript we used non-small cell lung carcinoma cell line – H1299. We have changed HT29 to H1299 (Line 215).
b) Indeed, we have used propidium iodide dye, thus Figure 1 legend was corrected (Line 292)
c) We have corrected this mistake.
2)
a) We have added the histological types of the used cell lines along with the information of their origin (Lines 96-107)
b,c) We have expanded the description of cellular assays used in this study (Lines 109-121). We have added information about the protocols, buffers, volumes and plates.
d) We have supplemented the description of PCR procedure with the additional information, including the protocol of amplification.
3) We have performed additional interference test of the compounds in the absence of cells. Indeed, MGL S3 and MGL tetani affect the results of colorimetric XTT test. However, as a) we have used 4 different cell lines in the study and for two of them we have observed significant reduction of cell viability determined by the XTT test and b) we have used the same volume of compounds to all the studied cell lines, we believe that these results are still reliable. We decided to reduce the amount of text devoted to the comparison of XTT and nuclei counting results. We think, that the presence of three different tests (XTT, nuclei counting, apoptosis) for the assessment of the sensitivity of cancer cells to MGL S3 and MGL tetani make our results more transparent and coherent.
4) To make our data presentation more structured and consistent, we have improved "Results" section and added the missing information.
All changes are tracked in the text, the most important changes are highlighted in yellow.
Reviewer 2 Report
Dear Authors,
I am recommending major corrections for your manuscript, please see below the point by point responses. The work has potential for publication, I can see it in lots of places throughout the manuscript. However, significant improvements are required in terms of presentation and quality.
Comment 1 (Line 45): “Most cancer cells are methionine dependent compared to non-malignant cells”.
Please provide few examples of the cancer cells
Comment 2 (Line 55): “There are several proposed mechanisms of methionine dependency of cancer cells”.
Brief description of mechanisms involved, currently the statement is very general.
Materials and method section should be listed before the result section.
Figure 1: Control samples showing healthy cells are missing.
Conclusion and future directions are missing. Those sections should be added. What’s the overall conclusion of study, have the aims been met, what are you doing next?
Author Response
Dear Reviewer,
First of all, we would like to thank you for the thoughtful and valuable comments.
We have significantly expanded "Introduction" section. Especially, we have extended the description of methionine dependency in cancer cells. We have supplemented text with the cell types and additional references. Lines 48-58. Also, we have added more information about the mechanisms of cancer cells dependency on methionine. Lines 59-66. We believe that these changes are crucial for the delivery of our idea and better understanding of the material by readers.
"Materials and methods" section was placed before the "Results". We have also added "Conclusions", where the main findings of the current research work as well as future directions were briefly described. Lines 515-528.
Additionally, we have corrected several parts in the "Results" section to make the experimental set up more clear and to highlight significance of the obtained results.
All changes are tracked in the text, the most important changes are highlighted in yellow.
Round 2
Reviewer 1 Report
The authors answered to all the reviewer's comments, added the missing information, and corrected the mistakes in the manuscript. Therefore, I recommend the publication of the new version of the manuscript.
Author Response
We would like to thank Reviewer for the valuable comments and helpful suggestions. We believe that they helped us to improve the quality of our manuscript.
Reviewer 2 Report
Dear authors,
The manuscript has been improved significantly in terms of presentation and quality. The points have been addressed well, therefore I am recommending acceptance.
Author Response
We appreciate the meaningful and thoughtful comments of Reviewer 2. We believe that they helped us to improve our manuscript.